# Vascular Contribution to Cerebral Waste Clearance Affected by Aging or Diabetes

**DOI:** 10.3390/diagnostics15081019

**Published:** 2025-04-16

**Authors:** Yimin Shen, Li Zhang, Guangliang Ding, Edward Boyd, Jasleen Kaur, Qingjiang Li, E. Mark Haacke, Jiani Hu, Quan Jiang

**Affiliations:** 1Department of Radiology, Wayne State University, Detroit, MI 48202, USA; ym_shen@wayne.edu (Y.S.); ehaacke@med.wayne.edu (E.M.H.); jhu@med.wayne.edu (J.H.); 2Department of Neurology, Henry Ford Health System, Detroit, MI 48202, USA; lzhang3@hfhs.org (L.Z.); gding1@hfhs.org (G.D.); eboyd3@hfhs.org (E.B.); jkaur3@hfhs.org (J.K.); qli1@hfhs.org (Q.L.); 3Department of Neurology, Michigan State University, East Lansing, MI 48824, USA; 4Department of Radiology, Michigan State University, East Lansing, MI 48824, USA; 5Department of Neurology, Wayne State University, Detroit, MI 48202, USA

**Keywords:** glymphatic system, cerebral vascular system, magnetic resonance imaging, susceptibility-weighted imaging, meningeal lymphatic vessels

## Abstract

**Background:** The brain’s vascular system has recently been shown to provide an important efflux pathway for cerebral waste clearance (CWC). However, little is known about the influence of aging or diabetes on the CWC. The aim of the current study is to investigate the vasculature contribution to CWC under aging and diabetic conditions. **Methods**: Male Wistar rats under aging and diabetic conditions were evaluated using dynamic intra-cisterna superparamagnetic iron oxide-enhanced susceptibility-weighted imaging (SPIO-SWI). Theoretical analysis of the expected signal intensity using SPIO-SWI was compared with the corresponding dynamic in vivo images. Quantitative susceptibility mapping (QSM) was used to evaluate the iron-based tracer concentration in the venous system. **Results**: Our data demonstrated that the theoretical analysis predicted the dynamic changes in the signal intensity after SPIO infusion. The distinct hyperintense signals due to the lower concentration of the SPIO over time in cerebrospinal fluid (CSF) and meningeal lymphatic (ML) vessels likely represented the CWC through various efflux pathways, including cerebral vascular and ML vessels. The QSM analysis further revealed reduced CWC from the vasculature in both the aged and diabetic groups compared to the younger group. **Conclusions:** Our results demonstrated that SPIO-SWI can quantitatively evaluate the CWC efflux contributions from cerebral vascular vessels under aging or diabetic conditions.

## 1. Introduction

Emerging data indicate that the cerebral waste clearance (CWC) plays an important role in neurological diseases [1,2,3,4,5,6,7,8,9]. The brain has long been considered to be devoid of a conventional lymphatic system. Studies in the last decade [1,2,3,4,5,10] have fundamentally altered the traditional model of cerebrospinal fluid (CSF) hydrodynamics and led to it being viewed as the glymphatic system responsible for the removal of interstitial fluid (ISF) and waste products out of the brain. The glymphatic system has been shown to be a sensitive biomarker for neurological diseases [1,2,3,4,5,6,8]. Although controversy exists in the efflux pathways [3,11,12,13], there seems to be solid consensus about the participation of the CSF pathways in CWC.

All cerebral blood vessels are surrounded by CSF [14]. Therefore, it might be no surprise to find that there is an interplay between the CSF and the vascular system that helps in CWC in the brain [15,16]. The traditional thought is that there is a major CWC pathway for non-specific substances through the vascular system via the arachnoid granulations in which brain waste enters the venous system through the sinus [17]. However, it is up for debate whether the arachnoid granulation is the only pathway for direct transfer of molecules from the CSF to the venous system [16,18,19]. Several studies [15,16,18,20,21,22] have demonstrated that there is rapid transport of the macromolecules into the blood stream from the brain parenchyma other than from the arachnoid granulations. However, identifying the contribution of the cerebral vascular system in CWC is a challenge due to the requirements of imaging tools that measure tracer influx and efflux quantitatively in the micro-vessels and their surroundings. In previous works, superparamagnetic iron oxide (SPIO)-enhanced susceptibility-weighted imaging (SWI, SPIO-SWI) [15,23] have been used to study this problem and it was found that the tracer can enter the parenchymal vascular system through the veins [15]. In the present study, we further investigate CWC in the parenchymal vascular system in both aging brains and in animal models of diabetes using SPIO-SWI.

## 2. Materials and Methods

All experimental procedures were conducted and performed in accordance with the guidelines of the National Institute of Health (NIH) for animal research under a protocol approved by the Institutional Animal Care and Use Committees of Henry Ford Hospital and Wayne State University, and experimental guidelines of ARRIVE (items 8, 10 to 13).

### 2.1. Animals and Experimental Procedures

Male Wistar rats (Charles River, Wilmington, MA, US) were used in the present study. Young adult (3–4 months, *n* = 8), aged adult (18–20 months, *n* = 10), and diabetic [24] (16–17 months, *n* = 5) rats were subjected to the identical experimental procedures, including surgical preparation for contrast agent administration via the cisterna magna, and subsequent MRI measurements. Diabetes was induced in middle aged (13 months of age) Wistar rats by intraperitoneal (IP) injection of 210 mg/kg of nicotinamide (NTM) and streptozotocin (STZ, 60 mg/kg). This approach has been demonstrated to produce noninsulin-dependent diabetic mellitus (DM) syndromes that resemble human type 2 diabetes [24,25,26]. The DM rats exhibited progression of cognitive decline starting at 2 months (2M) after STZ-NTM injection.

Surgery for catheter implantation into the cisterna magna was performed prior to performing MRI scans [27]. Briefly, the rats were initially anesthetized by inhalation of 3% isoflurane and then maintained in the range of 1.0–1.5% isoflurane with a mixture of N_2_O (70%) and O_2_ (30%) via a nose mask throughout the surgical period. Rectal temperature was strictly controlled at 37 ± 1 °C using a feedback-regulated water heating system. The head of the anesthetized rat was mounted in a stereotactic frame with care to permit spontaneous breathing. After the atlanto-occipital membrane was exposed using a midline dorsal neck incision, a polyethylene catheter (PE-10 tubing; Becton Dickinson, Cockeysville, MD, USA) filled with saline (10 µL) was inserted into the subarachnoid intra-cisterna magna (ICM) space via a small durotomy made with a 27-gauge needle. The outside part of catheter was fixed onto the occipital bone with superglue and the skin incision was closed around the catheter. The MRI tracer was superparamagnetic iron oxide (SPIO), specifically Ferumoxytol (Feraheme, AMAG Pharmaceuticals Inc., Waltham, MA, USA), a nonstoichiometric magnetite coated with polyglucose sorbitol carboxymethylether, with an overall colloidal particle size of 17–31 nm in diameter. The original iron concentration of Ferumoxytol was 30 mg/mL, which was saline diluted 20 times (1.5 mg/mL) for use. This tracer was administered into the cisterna magna via the implanted catheter at an optimized infusion rate of 1.67 μL/min [27] for 38 min using a 100-μL syringe (Hamilton Robotics, Reno, NV, USA), leading to a total infusion volume of about 63 μL CSF tracer or 180 µg Fe.

MR imaging was performed with a 7T system (Bruker–BioSpec AV4 Neo, Billerica, MA, USA) with ParaVision 360 (v2.0pl1) acquisition and processing software. A birdcage coil was used as the transmitter (2 channels) and a quadrature half-volume coil (4 channels) was used as the receiver. Anesthesia was initially induced by 3.5% isoflurane with medical air (1 L/min) for 3–5 min in an induction chamber attached to a scavenging F/air canister at the outlet. The animal with the catheter implantation was securely fixed on an MR-compatible cradle equipped with an adjustable nose cone for anesthesia and stereotaxic ear bars to immobilize the head. The nose cone provided both supply and return lines. A tooth bar through the nose cone was connected to the supply line to deliver the anesthesia gas and an active scavenging ventilation control (vacuum −40 mmHg) which reduced isoflurane exposure was connected on the nose cone as a return line. During image acquisition, anesthesia was maintained with 1.0–1.5% isoflurane (Piramal Inc., Bethlehem, PA, USA) and medical air (1 L/min), and the warm waterbed on the cradle was kept at 38 °C ± 1 °C.

The dynamic CSF tracer influx and clearance process was monitored using a 3D SWI scan. The imaging parameters were an echo time of 5.22 ms; repetition time of 35 ms; flip angle of 12°; field of-view of 19 × 19 × 30.72 mm^3^ (read, phase, slice) in a coronal orientation; matrix size of 384 × 192 × 192 (which yielded a resolution of 50 × 100 × 160 μm^3^); pixel bandwidth of 130 Hz; and an acquisition time of 30 min. The 30 min 3D SWI sequence was continuously acquired 4 times, and the 38-minute ICM infusion via implanted catheter was started after the first 3D scan, which served as pre-contrast baseline. After this, the 3D scan was repeated 4 times at longer intervals of 47 min with two other imaging sequences (a multi-slice 2D coronal *T*2 mapping (11 m 51 s) and a horizontal 3D *T*1-weighted imaging (4 m 38 s)) unrelated to this study. The post-contrast time was determined from the beginning of the infusion to the middle of each scan.

### 2.2. Imaging Data Analysis

MRI data were processed using in-house software Signal Processing in NMR (SPIN) (http://www.mrc.wayne.edu/, accessed on 24 June 2019). The gradient echo signal intensity (SI) is given by the FLASH equation [28]:(1)$$s\left(t\right)=\rho_{0}\frac{\sin\theta \left(1-e^{-TR/T1\left(t\right)}\right)}{\left(1-e^{-TR/T1\left(t\right)}\cos\theta \right)}e^{-\frac{TE}{T2^{*}\left(t\right)}}$$
where *ρ*_0_ is the spin density and *θ* is the flip angle. *T*1(*t*) and *T*2*(*t*) are expressed as(2)$$\frac{1}{T1\left(t\right)}=\;\frac{1}{T1\left(0\right)}+r_{1}c\left(t\right)$$
(3)$$\frac{1}{T2^{*}\left(t\right)}=\;\frac{1}{T2^{*}\left(0\right)}+r_{2}^{*}c\left(t\right)$$
where *c*(*t*) represents the contrast agent concentration, *T*1(0) and *T*2*(0) represent pre-contrast values of *T*1(*t*) and *T*2*(*t*), *r*_1_ and *r*_2_* represent the longitudinal [28,29,30] and effective transverse [23] relaxivities of the contrast agent. For Ferumoxytol, *r*_1_ = 3.1 (s^−1^ mM^−1^) and *r*_2_* = 106.8 (s^−1^ mM^−1^).

We simulated the effects of iron concentration on MRI signal intensity (SI) for gradient echo measurements used in dynamic tracer monitoring, based on Equations (1)–(3).

Quantitative susceptibility mapping (QSM) is a non-invasive, advanced MRI technique that measures the distribution of magnetic susceptibility in tissues [31]. Magnetic susceptibility is a property that describes how easily a substance is magnetized in an external magnetic field. QSM data were generated according to previous works [15] using SMART (Susceptibility Mapping and Phase Artifacts Removal Toolbox) software (v2.0) (MRI Institute for Biomedical Research, Bingham Farms, MI, USA). Briefly, the QSM reconstruction was performed using the brain extraction tool (BET) based on the magnitude images [32], and the three-dimensional phase unwrapping Sorting by Reliability following a Non-Continuous Path (3D-SRNCP) [33], followed by sophisticated harmonic artifact reduction for phase data (SHARP) for background field removal [34] with a kernel size of 8 and a regularization parameter of 0.05, and truncated k-space division with a filter threshold of 0.1. To improve susceptibility quantification, the iterative Susceptibility-Weighted Imaging and Mapping (iSWIM) method was used [35] with a vein threshold of 40 ppb and 8-pixel edge erosion for vessel structure extraction. In this case, a k-space threshold of 0.15, and 3 iterations were used. The susceptibility value is relative to that of the dominant background tissue in the brain which for the most part is water (−9.051 ppm) dominated [36].

The increase in susceptibility was directly proportional to the amount of SPIO tracer present [23]. A representative vein in brain parenchyma, the azygos internal cerebral vein (azicv), was chosen. Two-dimensional regions of interest (ROIs) were drawn manually on axial slices for the azicv, which were then combined into 3D-ROIs using SPIN software (Ver: 1.5.180723 (build:3265)) [15]. Vascular susceptibility (in parts per billion, ppb) was measured as an average over all voxels in the 3D-ROIs covering from the end of the dorsal septal vein to the great cerebral vein of Galen. The venous pixels were carefully selected to avoid touching the edges of the neighboring very small para-venous spaces. We report initial QSM values (without contrast agent) as the baseline for each group. To reduce the effect of potential variations in scale factor from scan to scan, the changes in percentage from baseline were used.

### 2.3. Statistical Analysis

Signal intensity of magnitude image was presented as mean (± standard error (SE)) of animals in a group. The susceptibility or their relative changes in percent (mean (±SE) for each group) are plotted as bar and line graphs, as appropriate. Within groups, two-tailed, paired sample *t*-tests were used to compare the baseline to post-tracer signal intensity measurements. Statistical significance was established at *p*-values less than the critical alpha value of 0.05 as the primary criterion and they were presented if any. Blue (the young group), red (the aged group) and green (the DM group) asterisks represent intra-group significant differences from pre-contrast baseline, respectively. Between groups, one-way analysis of variance (ANOVA, single factor) was conducted at each post-tracer time point. The F-values (ratio of variances between-groups to within-groups) being greater than the critical F-value, served as an alternative, were presented as well. The black asterisk (*) represents significant difference between groups. All statistics were performed using Microsoft Excel (Microsoft 365 MSO (Version 2503 Build 16.0.18623.20116) 64-bit).

## 3. Results

### 3.1. Visualization of Signal Intensity (SI) Changes After ICM Injection of Ferumoxytol

We first simulated the effects of iron concentration on MRI SI for the gradient echo measurement used in the dynamic tracer monitoring (Figure 1) based on Equations (1)–(3). The SI starting from the baseline (without contrast agent) increases when iron concentration is lower than ~1.2 mM with a maximum increase at ~0.3 mM. SI exhibits a monotonic decrease from baseline as the iron concentration increases from ~1.2 mM.

As predicted by the simulation results in Figure 1, both the decrease in SI for *T*2* shortening at high iron concentrations (>1.2 mM) and the increase in SI for *T*1 shortening at low iron concentrations (<1.2 mM) were detected after ICM administration of Ferumoxytol (Figure 2). The *T*2* effect (yellow arrows) and resulting signal loss appeared in the entire CSF/glymphatic pathway [24] shortly after injection since the concentration was high then. At later time points, as the concentration reduced, owing to CWC, the *T*1 shortening led to a higher signal (up white arrows) as shown in small portions in the subarachnoid spaces (SAS) although this effect disappeared gradually over time. For the *T*2* effect, the young group has the largest SI decrease in SAS (−39%, *p* = 0.01, at 45 min), compared to the aged group (−13%, at 75 min) and DM group (−5.4%, at 15 min). This is due to higher iron concentrations in young rats, indicating an effective glymphatic function compared to aged and DM groups, which allows the tracer to quickly distribute throughout the CSF in SAS. For the *T*1 shortening effect, the aged group has the largest increase in SI in SAS (46%, at 169 min, *p* = 0.001), compared to the young group (23%, at 210 min) and DM group (19.4%, at 169 min). The smaller SI increase in the young group over time caused by *T*1 shortening effect due to lower tracer concentration compared to initial, more pronounced SI decrease caused by *T*2* effect due to higher tracer concentration, suggests more effective glymphatic function and CWC through various efflux pathways including cerebral vascular and ML vessels in young rats compared to aged and DM rats.

In addition, we detected the dorsal meningeal lymphatic (ML) vessels adjacent to the superior sagittal sinus (SSS) in the parasagittal dural space after ICM infusion of Ferumoxytol (Figure 3) for CWC efflux pathway. In the young group, the SI in ML vessels gradually decreased until 75 min due to the *T*2* effect as the tracer concentration increased. The young group also showed a significant SI increase in ML vessels compared to pre-contrast baseline due to *T*1 shortening, peaking at 216 min (49%, *p* = 0.03), indicating effective tracer efflux through the ML vessels. However, for the aged group, SI in ML vessels slightly decreased at 15 min (−9%) and then showed a smaller increase peaking at 169 min (32%), while the DM group showed minimal signal changes within ±10%, indicating less tracer efflux through the ML vessels in both aged and DM groups compared to the young group. The detected ML vessels were discontinuous or piecewise around the SSS. The locations of the detected ML vessels were consistent with previous validation studies in human and non-human primates [37,38,39,40].

### 3.2. CSF Tracer Entry into the Parenchymal Veins Measured by QSM

The susceptibility of the CSF tracer was measured in the azicv. Figure 4 shows the QSM maps (a) and the corresponding susceptibilities (b) without contrast agent administration in young, aged and DM rats. The absolute values of susceptibility for the azicv were significantly lower in the young group (244 ± 33 ppb, *n* = 8) compared to the aged group (410 ± 28 ppb, *n* = 10, *p* = 0.003) and the DM group (351 ± 9 ppb, *n* = 5, *p* = 0.02). Smaller susceptibility in the vein indicated higher blood oxygen level for the young group.

Figure 5 shows the relative percentage changes in susceptibility from baseline of the azicv in the young, aged and DM groups after Ferumoxytol infusion. The relative susceptibility in the young group significantly increased at 45 min (24%, *p* = 0.003, blue asterisk), and then returned to relatively lower levels (<14%) afterwards, indicating a vascular contribution to CWC. Compared to the young group, the aged group showed less changes in the relative susceptibility; 7% increase at 15 min then back to relatively lower changes afterwards (<2%) at 122 min, then ~10% increase at 263 min. The DM groups showed less changes in relative susceptibility as well with ~6–8% increase from 15 min to 75 min then back to relatively lower changes afterwards (<4%). The aged and DM groups showed less changes in relative susceptibility without significant differences, indicating less vascular contribution to CWC in these groups when compared to the young rats. Between groups, the significant difference was only at 45 min time point (*p*-value of 0.004, F (7.56) > F critical of 3.52, denoted by a black asterisk).

## 4. Discussion

Accessing the vascular contribution to CWC has historically been a significant hurdle and unfortunately remains a challenge due to technical difficulties. In the current study, we demonstrated that the evolution of SI change after Ferumoxytol infusion in the ICM follows simulation results based on a theoretical analysis. A quick and large drop in temporal signal intensity in SAS (Figure 2) for the young rats indicates an effective glymphatic function compared to the aged and DM rats. The distinct hyperintense signals due to the lower concentration of the SPIO over time in cerebrospinal fluid (CSF) and meningeal lymphatic (ML) vessels likely represented the CWC through various efflux pathways, including cerebral vascular and ML vessels. Furthermore, our QSM data demonstrated reduced CWC from the vasculature in both aged and DM groups compared to the young group.

The current study is the first investigation of brain vascular contributions to CWC with respect to age and DM using SPIO-SWI, although aging and DM effects on the impairment of glymphatic pathways for CWC have been investigated previously [24,37,41,42]. Due to the limitation of low penetration and narrow field of view in optical fluorescent imaging, Gadolinium-based dynamic MRI (Gad-MRI) with intrathecal infusion of contrast agent has been the most reliable means for whole brain CWC assessment. However, previous investigations of CWC using Gd-MRI only focused on glymphatic and periaxonal/perineural routes. Moreover, the non-invasive DTI-ALPS technique has been proposed as an indirect method to measure the glymphatic system [43], although subsequent studies have highlighted its limitations in accurately reflecting the glymphatic function [44,45]. It has been reported that the DTI-ALPS index decreased in patients with type-2 DM as compared to healthy controls [46], which is consistent with the impaired CWC observed in our study. The vascular contribution to the efflux of CWC has traditionally focused on the arachnoid villi/granulation pathway outside of the brain parenchyma, which comes from a cadaver brain study conducted over 100 years ago under non-physiologically high CSF pressures [17]. Debate about direct transfer of molecules from the CSF to the venous system via arachnoid granulations has been considered [16,18]. Several studies have demonstrated that there is rapid transport of the macromolecules into the blood stream from the brain parenchyma other than from the arachnoid granulations [16,18,20,21,22] and cerebral venous vessels may play a substantial role in CWC [15]. The current study demonstrated that the reduced CWC observed in aging or DM conditions may result not only from impaired glymphatic, ML and periaxonal pathways but also, in part, from compromised cerebral vascular efflux.

Our results demonstrated that SPIO-SWI has advantages in detecting the vascular contribution to CWC and, potentially, detecting the cerebral waste efflux through the ML system. Currently, we face technical challenges for the in vivo detection and characterization of the microvascular abnormalities in the brain [47]. Although 3D SWI venography is an important clinical tool to image the venous system that is independent of blood flow [23], its current imaging resolution is limited by the partial volume effect, making it unable to detect smaller veins. To overcome the above-mentioned limitations of the current technology, SPIO-SWI has shown significant potential to achieve resolutions as fine as ~10 μm in order to visualize and quantify micro-vascular changes in vivo in animals [15]. Our previous studies demonstrated that SPIO-SWI with ICM infusion of SPIO can simultaneously distinguish arteries, veins and their corresponding peri-vascular spaces, as well as identify glymphatic pathways for dynamic input and clearance of tracers [15]. The results also suggested the direct participation of the parenchymal vascular system in CWC, especially from the cerebral veins [15].

In the current study, we also demonstrated the possibility of identifying ML efflux using the hyperintense SI due to the low concentration (<1.2 mM) of SPIO in addition to the quantitative evaluation of SPIO concentration (>1.2 mM) with hypointense signal due to the high concentrations as demonstrated in Figure 2 and Figure 3. Although we were able to detect ML vessels adjacent to the superior sagittal sinus (SSS) in the peri-SSS space at the same anatomic location as in previous validation studies in human and non-human primates [37,38,39,40], the validation of MRI-detected ML vessels in those works was performed using ex vivo immunohistochemical staining methods which may involve some degree of error [40,48,49]. The best validation of MRI-detected ML vessels is to employ transgenic lymphatic reporter animals. Currently, there is still conflicting information about the CWC in the dorsal ML vessels and it is an area of active investigation [37,38,39,40,50,51,52].

The mechanisms underlying the reduced CWC observed in the vascular system under aging or DM conditions are unclear. One might consider changes in the CSF dynamics, physiological or pathological conditions as sources of CWC changes. For example, changes in CSF tracer flow dynamics such as reduced infusion and clearance rates in the aging or diabetic brain could result from decreased CSF production [53,54,55], increased CSF outflow resistance [56,57], reduced lymphatic CSF transport [58,59] or dural lymphatic dysfunction [60]. The suppression of brain-wide perivascular transport may be, in part, attributed to the aging- or diabetes-dependent alterations in the cerebral vascular system, including the decline in vascular pulsatility [42,61], increase in vessel stiffness [8,62], loss of perivascular AQP4 polarization [24,42,63], abnormalities in perivascular space [24,64,65], decrease in microvascular density [66,67] and neurovascular uncoupling [68,69]. However, the mechanism-related causes of the reduced CWC through the vascular system in aging or diabetic conditions require further investigation.

Our data exhibited increased susceptibilities of venous blood in the aged and diabetic groups even before Ferumoxytol infusion compared to the young group. The increased susceptibilities in the aged and diabetic groups are likely caused by increased deoxyhemoglobin (reduced oxygen saturation) in the venous blood [70,71,72,73,74]. QSM has been used to measure venous oxygen saturation in the determination of predictors for progressive ischemic regions in urgent care settings [75,76]. Previous investigations demonstrated reduced oxygen saturation with increased magnetic susceptibility in aging or diabetic conditions consistent with our results [70,71,72,73,74].

Moreover, both aging and diabetes increase the risk of heart failure, and biomarkers such as Troponin T (TnT) and NT-proBNP are established indicators of myocardial damage [77]. A recent study also demonstrated the dysregulation of brain fluid dynamics after myocardial infarction, indicating the role of impaired CWC mechanisms in heart failure [78]. These findings highlight the connection between cardiovascular and neurological health. Therefore, future investigations are needed to measure CWC to clarify its contribution to both cardiovascular and neurological diseases.

In summary, SPIO-SWI combined with ICM infusion of Ferumoxytol could provide quantitative measurements of impaired CWC through cerebral vasculature under aging or diabetic conditions. This technique can also effectively detect ML vessels, providing insight into other efflux pathways involved in CWC. Further investigations are required to validate the ML pathways and solidify its relationship with the glymphatic system, both in healthy individuals and those with neurological diseases.

## Figures and Tables

**Figure 1 diagnostics-15-01019-f001:**
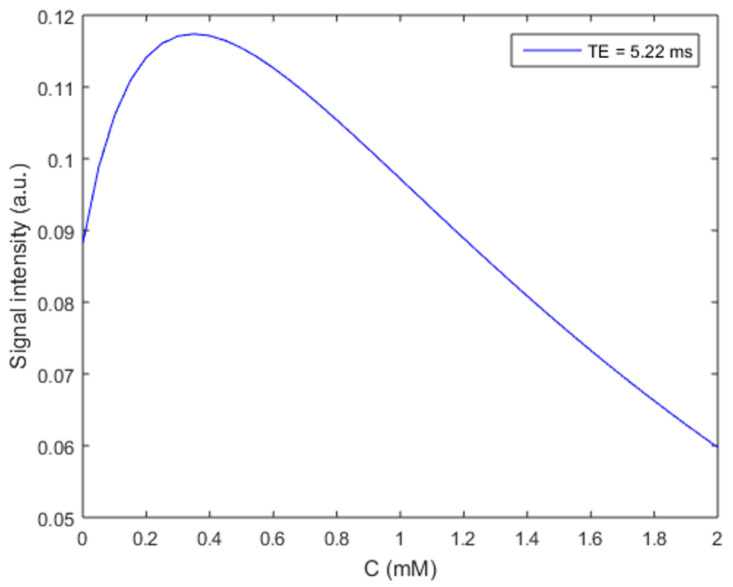
Gradient echo MRI signal intensity changes with iron contrast agent (Ferumoxytol) concentration. The effective transverse (*r*2* = 106.8 s^−1^ mM^−1^) [24] and longitudinal (*r*1 = 3.1 s^−1^ mM^−1^) relaxivities of Ferumoxytol at 7T were used along with *T*1_0_ = 2 s, *T*2_0_* = 50 ms, *TR* = 35 ms, Flip angle = 12°, *ρ*_0_ = 1, and *TE* = 5.22 ms to generate this curve based on Equations (1)–(3).

**Figure 2 diagnostics-15-01019-f002:**
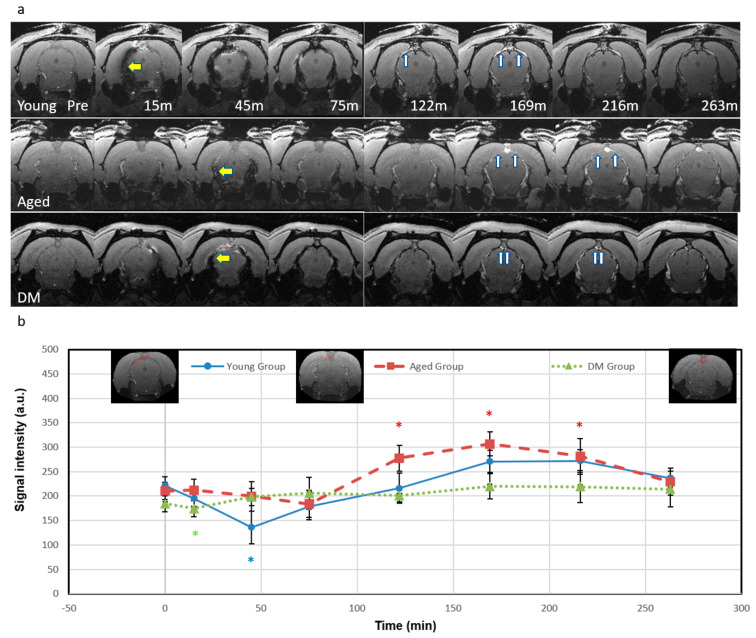
Temporal signal intensity of magnitude images. (**a**) Temporal axial magnitude images (bregma −7.2 mm) for representative rats from the young group (**top** row), the aged group (**middle** row), and the DM group (**bottom** row) show *T*2* and *T*1 effects in the representative rats. The *T*2* effect shows low signal (yellow arrows) first and the *T*1 effect shows high signal (white arrows) as the concentration reduces. (**b**) Temporal plots of signal intensity in the subarachnoid space (mean (SE)). For the young group, the low signal was detected during 15 to 75 min and reached lowest at 45 min (−39%, *p* = 0.01), then signal started to rise at 122 min, increased at 169 min (22%) and 210 min (23%), and finally dimmed. For the aged group, the signal showed a slight decrease at 45 min (−5.3%) and reached its lowest point at 75 min (−12.5%). It then increased significantly, rising at 122 min (32%, *p* = 0.005), peaked at 169 min (46%, *p* = 0.001) to 210 min (34%, *p* = 0.02), and finally dimmed. For DM group, the signal showed a slight initial decrease at 15 min (−5.4%, *p* = 0.03), and the signal started to increase at 45 min and peaked at 169 min (19%). There were no significant differences in signals within the SAS between the groups at any time point and therefore there is no black asterisk (*) in this figure. Blue (the young group), red (the aged group) and green (the DM group) asterisks represent intra-group significant differences from pre-contrast baseline, respectively. The insets in Figure 2b show the regions of interest.

**Figure 3 diagnostics-15-01019-f003:**
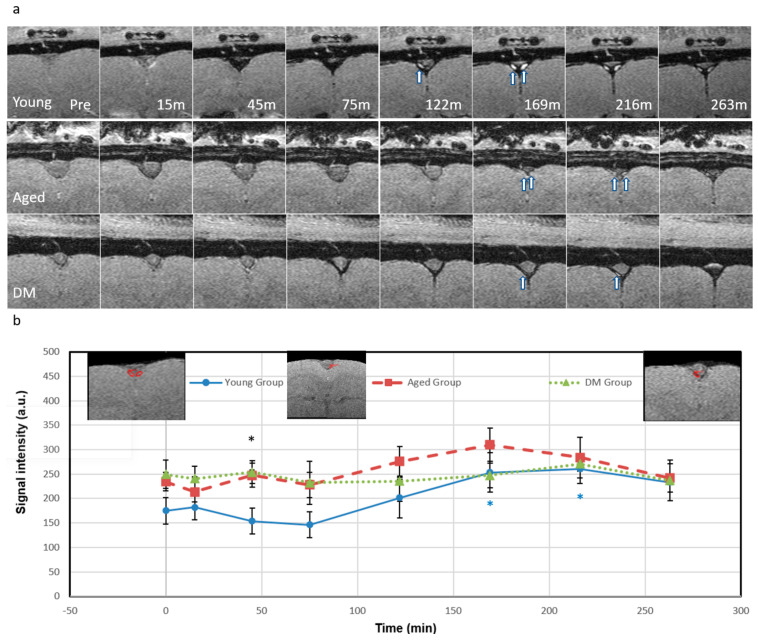
Detection of meningeal lymphatic vessels (ML) in the parasagittal dural space. (**a**) Temporal axial magnitude images (bregma −4.2 mm) for young (**top** row), aged (**middle** row), and DM (**bottom** row) groups showing *T*2* and *T*1 effects (the same rats in Figure 2); (**b**) The meningeal lymphatic vessel signal intensity temporal plots. The arrows indicate the ML vessels adjacent to the superior sagittal sinus in the parasagittal dural space consistent with published results [40,41,42,43]. For the young group, the ML signal gradually decreased (up to 75 min) as tracer concentration increased resulting in the *T*2* effect. After 122 min, as the tracer concentration decreased, the signal started to increase showing the *T*1 shortening effect due to cerebral waste clearance. The signal significantly increased at 169 min (45%, *p* = 0.04) and peaked at 216 min (49%, *p* = 0.03). Then, the signal started to drop back. For the aged group, the ML signal slightly decreased at 15 min (−9%) and 75 min (−2.7%), then started to increase at 122 min (18%), peaked at 169 min (32%), and then finally dropped back (3.2%). For the DM group, the ML signal had a small variation within ±10%. The significant difference between the three groups was only at 45-minute time point (*p*-value of 0.032, F (4.09) > F critical of 3.49) denoted by a black asterisk. At the time point, the young group showed high tracer concentration (represented by low signal due to *T*2* effect) while the aged and DM groups showed low tracer concentrations. Black asterisk represents significant difference between groups. Blue (the young group) asterisk represents intra-group significant differences from pre-contrast baseline. The insets in Figure 3b show the regions of interest.

**Figure 4 diagnostics-15-01019-f004:**
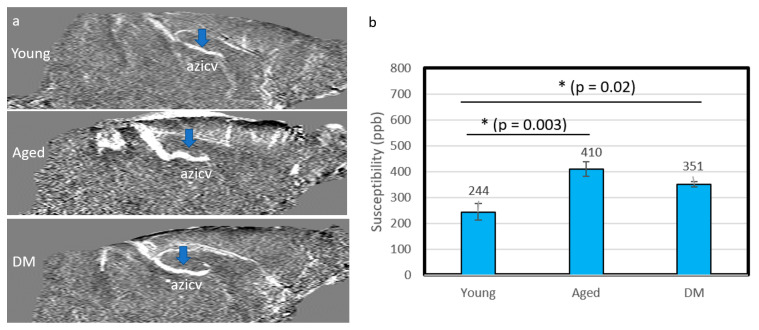
QSM in the internal cerebral vein (azicv, blue arrows) for the young, aged and DM groups without contrast agent. (**a**) Mid-sagittal view of QSM. (**b**) The corresponding susceptibility values in the three groups. The susceptibility of the young group is significantly lower than that in the aged and DM groups, indicating lower levels of deoxyhemoglobin. Black asterisk represents significant difference between groups.

**Figure 5 diagnostics-15-01019-f005:**
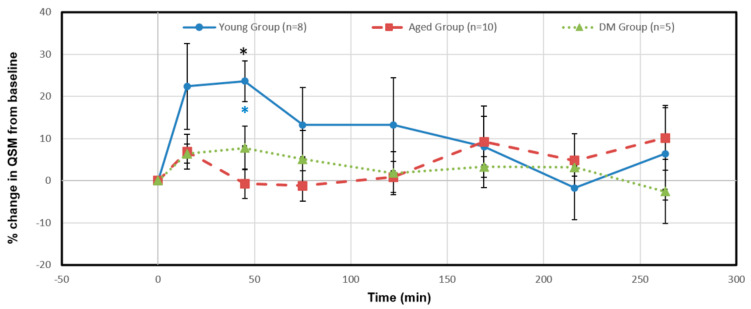
Temporal percentage changes in QSM of the vein (azicv) for the young (blue line), aged (red broken line) and DM (green dot line) groups after Ferumoxytol infusion via the intra cisterna magna. The relative susceptibility in the young group significantly increased at 45 min (24%, *p* = 0.003). The significant difference between the three groups was only at the 45 min time point (*p*-value of 0.004, F (7.56) > F critical of 3.52). Black asterisk represents significant difference between groups. Blue (the young group) asterisk represents intra-group significant differences from pre-contrast baseline.

## Data Availability

The raw data supporting the conclusions of this article will be made available by the corresponding author on request, considering the extensive size of the dataset. Minimal dataset is represented in the figures and data plots included in the manuscript.

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
