# Peer review of "Vascular Contribution to Cerebral Waste Clearance Affected by Aging or Diabetes"

_diagnostics, 2025, doi:10.3390/diagnostics15081019_

Round 1
Reviewer 1 Report
Comments and Suggestions for Authors
This study investigates the impact of aging and diabetes on cerebral waste clearance (CWC) in rats, using superparamagnetic iron oxide enhanced susceptibility-weighted imaging (SPIO-SWI). The research focuses on the brain's vascular system's role in removing waste, which is essential for neurological health. The findings indicate that both aging and diabetes impair CWC, with reduced clearance observed in the vasculature of aged and diabetic rats compared to younger ones. SPIO-SWI proves valuable for quantitatively assessing CWC contributions from cerebral vascular vessels under these conditions and detecting meningeal lymphatic vessels. The study suggests that impaired CWC in aging and diabetes may stem from compromised cerebral vascular efflux, alongside glymphatic and other pathway impairments.
I found this manuscript sound, well-written, and well-organized. The authors paid attention to each section carefully and wrote properly.
I have only some suggestions to improve the discussion:
1. As you know, DTI-ALPS, developed by Taoka et al. in 2017, works on the glymphatic system in humans in perivascular spaces. I highly recommend that the authors mention results from diabetic individuals to align your work and improve your text, such as PMCID: PMC11150564.
2. Regardless of SWI and other techniques, QSM has been previously documented in humans and animals. I think you missed important papers that could make better sense of your text on using non-invasive techniques such as QSM.
Author Response
Thank you very much for taking the time to review our manuscript titled “Vascular Contribution to Cerebral Waste Clearance Affected by Aging or Diabetes”. Please find the detailed responses below and the corresponding revisions in track changes in the revised manuscript.
Comment 1: As you know, DTI-ALPS, developed by Taoka et al. in 2017, works on the glymphatic system in humans in perivascular spaces. I highly recommend that the authors mention results from diabetic individuals to align your work and improve your text, such as PMCID: PMC11150564.
Response 1: Thank you for your suggestion. We have added the diabetes study in our revised manuscript in lines 310-315.
Comment 2: Regardless of SWI and other techniques, QSM has been previously documented in humans and animals. I think you missed important papers that could make better sense of your text on using non-invasive techniques such as QSM.
Response 2: Some QSM studies are discussed in lines 365-371. We have added QSM-related information in lines 138-141.
Reviewer 2 Report
Comments and Suggestions for Authors
Dear authors,
Thank you for the opportunity to review your manuscript entitled "Vascular Contribution to Cerebral Waste Clearance Affected by "Aging or Diabetes".
Abstract, title and references. The aim of the study is clear. The title is informative and relevant. The references are relevant, recent, and referenced correctly.
Introduction. It is clear what is already known about this topic. The research question is clearly outlined.
Methods. The process of subject selection is clear. The variables are defined and measured appropriately. The study methods are valid and reliable. There is enough detail in order to replicate the study.
Results and Discussion. The results are discussed from multiple angles and placed into context without being overinterpreted. The conclusions answer the aims of the study.
The conclusions supported by references and results.
The limitations of the study are opportunities to inform future research. Overall.
The study design was appropriate to answer the aim. The manuscript is well written and a stimulus for the readership.
Minor revisions: An interesting model for assessing cerebral caste clearance in the course of aging and diabetes has been described. As we know, the risk of developing heart failure increases with age and in the course of diabetes. Troponin T and NTproBNP (1) are recognized biomarkers that describe well the progression of myocardial damage.
Was the correlation between TnT and NT-proBNP and the degree of CWC impairment assessed in the presented study?
It is worth referring to the issue of heart failure and CWC impairment (1) in the discussion. Reference 1. Doi: 10.33963/v.phj.99553
Best regards
Author Response
Thank you very much for taking the time to review our manuscript titled “Vascular Contribution to Cerebral Waste Clearance Affected by Aging or Diabetes”. Please find the detailed responses below and the corresponding revisions in track changes in the revised manuscript.
Comment 1: An interesting model for assessing cerebral caste clearance in the course of aging and diabetes has been described. As we know, the risk of developing heart failure increases with age and in the course of diabetes. Troponin T and NTproBNP (1) are recognized biomarkers that describe well the progression of myocardial damage. Was the correlation between TnT and NT-proBNP and the degree of CWC impairment assessed in the presented study?
Response 1: Thank you for your insightful question. Although the risk of developing heart failure increases with age and diabetes, and Troponin T (TnT) and NT-proBNP are established biomarkers of myocardial damage, in our current study we did not assess the correlation between these cardiac biomarkers and the degree of cerebral waste clearance (CWC) impairment. However, this is certainly an interesting area for future investigation.
Comment 2: It is worth referring to the issue of heart failure and CWC impairment (1) in the discussion. Reference 1. Doi: 10.33963/v.phj.99553
Response 2: Based on your suggestions, we have added the relevant studies in lines 372-378.
Reviewer 3 Report
Comments and Suggestions for Authors
This manuscript presents a investigation into the cerebral waste clearance (CWC) system, utilizing SPIO-enhanced susceptibility-weighted imaging (SPIO-SWI) and quantitative susceptibility mapping (QSM) to noninvasively evaluate cerebral clearance in animal models. The study addresses an important gap in the field of neuroimaging and glymphatic research, and the methodology is technically sound.
Author Response
Comment 1: This manuscript presents a investigation into the cerebral waste clearance (CWC) system, utilizing SPIO-enhanced susceptibility-weighted imaging (SPIO-SWI) and quantitative susceptibility mapping (QSM) to noninvasively evaluate cerebral clearance in animal models. The study addresses an important gap in the field of neuroimaging and glymphatic research, and the methodology is technically sound.
Response 1: Thank you very much for taking the time to review our manuscript titled “Vascular Contribution to Cerebral Waste Clearance Affected by Aging or Diabetes”.